# Overexpression of GATA5 Inhibits Prostate Cancer Progression by Regulating PLAGL2 via the FAK/PI3K/AKT Pathway

**DOI:** 10.3390/cancers14092074

**Published:** 2022-04-21

**Authors:** Qinghua Wang, Zelin Liu, Guanzhong Zhai, Xi Yu, Shuai Ke, Haoren Shao, Jia Guo

**Affiliations:** Department of Urology, Renmin Hospital of Wuhan University, 99 Zhangzhidong Road, Wuhan 430060, China; wqh1996@whu.edu.cn (Q.W.); 2019283020192@whu.edu.cn (Z.L.); 2019283020168@whu.edu.cn (G.Z.); yuxi2018@whu.edu.cn (X.Y.); 2020283020212@whu.edu.cn (S.K.); 2017302180212@whu.edu.cn (H.S.)

**Keywords:** GATA5, prostate cancer, PLAGL2, proliferation, metastasis, EMT, FAK/PI3K/AKT pathway

## Abstract

**Simple Summary:**

Prostate cancer (PCa) has the highest incidence of malignant tumors and is the second-ranked tumor-causing death of men. GATA binding protein 5 (GATA5) belongs to the GATA gene family and we found that GATA5 was downregulated in PCa tissues, but the function of GATA5 in PCa remains elusive. We found overexpression GATA5 inhibited tumor proliferation, migration, invasion and the process of epithelial–mesenchymal transition (EMT), and upregulation of GATA5 promoted PCa cell apoptosis. In addition, we disclosed that GATA5 could interact with pleomorphic adenoma gene-like-2 (PLAGL2) to regulate PCa cell growth via FAK/PI3K/AKT signaling pathway. Hence, these findings suggested that GATA5 could serve as a new therapeutic target in the future.

**Abstract:**

Background: Prostate cancer (PCa) is a malignancy with high incidence and the principal cause of cancer deaths in men. GATA binding protein 5 (GATA5) belongs to the GATA gene family. GATA5 has a close association with carcinogenesis, but the role of GATA5 in PCa remains poorly understood. The aim of our present study was to probe into the effect of GATA5 on PCa progression and to elucidate the involved mechanism. Methods: The expression of GATA5 was detected in both PCa samples and PCa cell lines. GATA5 overexpression, PLAGL2 knockdown, and overexpression cell models were generated, then Western blotting experiments were utilized to validate the efficiency of transfection. The effects of GATA5 on PCa cell proliferation, metastasis, apoptosis, cell cycle progression, and EMT were detected in vitro or in vivo. Furthermore, the mechanism by which GATA5 inhibits prostate cancer progression through regulating PLAGL2 via the FAK/PI3K/AKT pathway was also explored. Results: GATA5 expression was downregulated in PCa samples and cell lines. GATA5 overexpression inhibited PCa cell proliferation and metastasis but increased the rate of apoptosis. In addition, we confirmed that GATA5 inhibited prostate cancer progression, including EMT, by regulating PLAGL2 via the FAK/PI3K/AKT pathway. Conclusion: We demonstrated that GATA5, as a tumor suppressor in PCa, inhibits PCa progression by regulating PLAGL2. These results showed that the GATA5/PLAGL2/FAK/PI3K/AKT pathway may become a new therapeutic direction for the treatment of PCa.

## 1. Introduction

Prostate cancer (PCa) has become the most common malignancy of men in the USA, and cancer-related deaths due to PCa are only fewer than those due to lung cancer [1]. The incidence of prostate cancer in China is lower than that in the West, but it is rising rapidly as life expectancy increases and early screening and detection technologies progress [2]. Many patients with early prostate cancer have no obvious symptoms due to insidious onset [3]. Patients will experience symptoms, such as urinary tract obstruction, sexual dysfunction, or even bone pain, as the disease progresses [4]. Most patients, who are at the first diagnosis, are at an advanced stage, and the patient’s quality of life in the future is also poor. Currently, the treatment of early prostate cancer is mainly surgery and androgen deprivation therapy [5,6]. Most prostate cancer patients who are in the early stage are sensitive to androgen deprivation therapy (ADT); however, they inevitably develop castration-resistant PCa (CRPC), and the five-year survival rate of patients is not optimistic [7]. Therefore, it is vital to further study the pathogenesis of prostate cancer and find new molecular markers.

GATA binding protein 5 (GATA5), located on chromosome 20q13.33 with 8 exons, belongs to the class of transcription factors containing 2 zinc fingers. GATA5 is a member of the GATA gene family, which includes GATA1, GATA2, GATA3, GATA4, GATA5, and GATA6 [8]. GATA1/2/3 play an essential role in the differentiation of ectoderm and mesoderm, while GATA4/5/6 are vital for the differentiation and development of the endoderm [9,10]. It has been reported that altered expression of GATA5 is relevant to cardiovascular diseases, including arrhythmia, hypertension, and congenital heart disease [11,12,13]. Genetic and epigenetic data have shown that aberrant methylation of GATA5 promotes the development of gastric diseases [14]. In addition, abnormal expression of GATA5 is also involved in the progression of various tumors. The level of GATA5 protein in colorectal cancer tissues is clearly lower than in normal tissues due to DNA promoter methylation; in contrast, overexpression of GATA5 significantly inhibits the progression of colorectal cancer cells [15,16]. GATA5 CpG island methylation may become a potential biomarker for the detection and prognosis of renal cell carcinoma [17,18]. Loss of GATA5 is involved in the development of hepatocellular carcinoma [19,20]. Overall, GATA5 is closely associated with carcinogenesis, but the function of GATA5 in PCa is poorly understood.

Analysis of the STRING databases showed that GATA5 may interact with pleomorphic adenoma gene-like-2 (PLAGL2), which is a member of the PLAG family. PLAGL2 plays a carcinogenic role and is related to the development of various cancers. In gastric cancer, overexpression of PLAGL2 facilitates the proliferation and migration of gastric cancer cells and contributes to the process of epithelial–mesenchymal transition (EMT) via the USP37-Snail1 axis [21]. PLAGL2 also facilitates the colorectal cancer metastasis through relying on ZEB1 [22]. Moreover, PLAGL2 overexpression plays a vital role in hepatocellular carcinoma and induces tumor metastasis via EGFR-HIF-1/2α signaling, and cell lines with high PLAGL2 expression have erlotinib resistance, which can be a biomarker for erlotinib therapy [23]. We have previously demonstrated that PLAGL2 is overexpressed in PCa tissues and can serve as an independent factor for the prognosis [24]. However, the precise mechanism by which PLAGL2 acts as an oncogene in PCa is still unclear.

KEGG pathway analysis has indicated that GATA5 may regulate PCa progression through the focal adhesion kinase (FAK) signaling pathway, which participates in multiple fundamental processes, including angiogenesis, proliferation, and metastasis, as well as EMT in various cancer cells [25]. Previous studies have reported that aberrant phosphorylation of FAK promotes PCa progression, while inhibition of FAK enhances the response to chemotherapy [26,27]. Protein kinase B (AKT) and phosphatidylinositol-3-kinase (PI3K) are the downstream genes of FAK [28]. The present study aimed to identify whether GATA5 inhibits PCa cells through regulating PLAGL2 via the FAK/PI3K/AKT pathway.

In the present study, we demonstrated that GATA5 was downregulated in both PCa samples and cell lines. Additionally, GATA5 overexpression substantially suppressed PCa cell proliferation, apoptosis, and metastasis, and GATA5 overexpression significantly inhibited PLAGL2 expression and FAK activity. In conclusion, our findings suggested that the downregulation of GATA5 is critical for the progression of PCa and that GATA5 may be a potential biomarker for the diagnosis and treatment of PCa.

## 2. Materials and Methods

### 2.1. Patient Specimens and Cell Culture

All PCa tissues and adjacent cancer tissues were collected from Renmin Hospital of Wuhan University (Wuhan, China) between January 2018 and January 2021, and all the patients had not undergone chemotherapy or radiation therapy before the operation. All tissues were placed in liquid nitrogen or 4% paraformaldehyde after being isolated.

PCa cell lines (PC3 and DU145) and normal prostate epithelial cells (RWPE-1) were acquired from the Cell Bank of the Chinese Academy of Sciences (Shanghai, China). All PCa cells were cultured in RPMI-1640 medium (HyClone, Logan, UT, USA) with 1% sodium penicillin G/streptomycin sulfate and 10% fetal bovine serum (FBS) (Gibco, Waltham, MA, USA), and RWPE-1 cells were incubated in K-SFM medium (Gibco, Waltham, MA, USA) containing 10% FBS. A humidified chamber containing 5% CO_2_ was used to incubate all cells at 37 °C.

### 2.2. Western Blotting

Cell or tissue samples were lysed in RIPA lysis buffer on the ice (Beyotime, Shanghai, China) with 1% phosphatase inhibitor (Servicebio, Wuhan, China), then the lysates were centrifuged right away at 13,000 rpm for 15 min after being incubated on ice for 30 min. The supernatant from each sample was then collected, and the BCA assay was used to detect the concentration of sample protein. After the addition of 25% volume loading buffer, all samples were boiled for 15 min at 100 °C. All protein samples were preserved at −80 or −20 °C for different types of storage. Equivalent protein from each sample was separated by 12% SDS-PAGE for 2 h at 80 V, and then protein was transferred to PVDF membranes (Millipore, NJ, USA) at a current of 200 mA. The membranes were then blocked with protein-free rapid blocking buffer (Epizyme, Shanghai, China) for 15 min at room temperature. The membranes were then incubated with various primary antibodies overnight at 4 °C, followed by incubation with a secondary antibody for 70 min. Specific bands were tested with an ECL kit (Beyotime, Shanghai, China) and analyzed with ImageJ software. The primary antibodies are listed in Table 1.

### 2.3. Immunoprecipitation (IP)

PC3 or DU145 cells were lysed in RIPA lysis buffer on ice with 1% Cocktail (Servicebio, Wuhan, China) for 30 min, and the lysate was centrifuged at 10,000–14,000× *g* for 5 min at 4 °C. Next, IgG (Proteintech, Wuhan, China) or primary antibody was added into the suspension at 4 °C overnight, and the Protein A + G beads (Beyotime, Shanghai, China) were added to the mixture for incubation. After centrifugation of the beads, the supernatant was removed and washed three times with the inhibitor-containing lysate. SDS-PAGE Sample Loading Buffer (1X) was then added to the samples and heated at 95 °C for 5 min. Finally, the supernatant was taken for Western blotting after separation.

### 2.4. Quantitative Real-Time PCR

Total RNA from cell or tissue was extracted using TRI Reagent (Absin, Shanghai, China). cDNA was synthesized by employing the PrimeScript™ RT Reagent Kit (TaKaRa, Bijing, China) in accordance with the manufacturer’s instructions. TB Green^®^ Premix Ex Taq™ II (TaKaRa, Bijing, China) was utilized for the qRT-PCR assay. The primer sequences in our study were as follows (Sangon Biotech, Shanghai, China):

GAPDH forward, 5′-TGACTTCAACAGCGACACCCA-3′; GAPDH reverse, 5′-CACCCTGTTGCTGTAGCCAAA-3′.

GATA5 forward, 5′-CATCACAGACTTACGCACTTGTTTGG-3′; GATA5 reverse, 5′-GGATGGGTCAGGACAGGGCTAC-3′.

### 2.5. Cell Transfection Assay

To construct a GATA5 overexpression cell model, the GATA5 overexpression plasmid (pcDNA3.1-hGATA5) was constructed (Viraltherapy, Wuhan, China). Transient transfection was implemented using Lipofectamine 2000 (Invitrogen, Waltham, MA, USA) when the cell density was 50–70%. For the generation of si-PLAGL2 cells, a small-interfering RNA targeting PLAGL2 (RiboBio, Guangzhou, China) was synthesized and transfected into cells, as described above. The PLAGL2 overexpression cells were constructed as above (Viraltherapy, Wuhan, China). The efficiency of transfection was measured by Western blotting analysis.

### 2.6. Cell Proliferation Assay

Cells (2000 cells/well) were seeded in 96-well plates (Corning, Corning, NY, USA). At the appointed time, the medium with 10% CCK-8 (Biosharp, Beijing, China) was added to each well. After incubation for 2 h at 37 °C, each well was detected at 450 nm and the data were analyzed to assess the cell proliferation ability. In addition, an EdU experiment (RiboBio, Guangzhou, China) was also utilized to explore cell proliferation. Cells (1 × 10^4^ cells/well) were plated into 96-well plates. Cells were then incubated with 50 µM of EdU for 2 h at 37 °C. Cells of each well were then fixed, permeabilized, and stained with Apollo, and the cell nucleus was stained. Cell proliferation was assessed by counting the ratio of red-light cells labeled with EdU.

### 2.7. Colony Formation Assay

Cells (600 cells) were seeded into 6-well plates (Corning, Corning, NY, USA) and the status of cells was observed and the medium was changed every 2–3 days. Cells were fixed with paraformaldehyde for 20 min after culturing for 2 weeks, and then crystal violet was used to stain the colony. Visible colonies (at least 50 cells) were counted and the data were analyzed to compare colony formation capacity.

### 2.8. Wound-Healing Assay

Cells were cultured in 6-well plates and scratched with a standard 200 μL sterilized pipette when the cells reached approximately 90% confluence. Each group was then cultured with 2% FBS medium. Then, 0 and 24 h after scratching, images were acquired.

### 2.9. Transwell Invasion Assay

Matrigel (Corning, Corning, NY, USA) was placed on ice and diluted with PBS, and the 24-well plate with chambers (Corning, Corning, NY, USA) was placed in the incubator overnight after adding 70 μL of prepared Matrigel. The next day, DU145 cells (3 × 10^4^) or PC3 cells (5 × 10^4^) suspended in 200 μL of medium with 2% FBS were seeded into the upper chamber. Then, 700 μL of medium with 10% FBS was added to the lower chamber. The chamber was washed three times with PBS after 48 h of incubation. The chambers were then fixed in 4% paraformaldehyde and stained with crystal violet, and the cells in the inner layer were gently wiped off with a cotton swab. After the chambers were dried, the cells that invaded were counted under a microscope.

### 2.10. Cell Apoptosis and Cell Cycle

Cells (10,000 cells/mL) were plated in a 6-well plate and placed in an incubator containing 5% CO_2_ overnight at 37 °C. Cells were then collected, and precooled PBS was used to wash them, and then they were centrifuged. Next, 300 μL of binding buffer was added to the cell pellet for resuspension. Annexin V-FITC was then added followed by incubation in the dark for 10 min (BD Biosciences, NJ, USA). Cell apoptosis was detected after adding PI. For cell cycle analysis, all the cells were collected, centrifuged, washed three times with precooled PBS, fixed with precooled 70% ethanol, and incubated overnight at 4 °C. PI was added after washing, and cells were then resuspended for cell cycle detection.

### 2.11. Immunohistochemistry (IHC)

IHC was performed to assess the expression level of protein in tissues. All tissues were cut into about 4 μm-thick sections after being embedded in paraffin. The sections were then dewaxed, dehydrated, and boiled for antigen retrieval. After blocking the sections with hydrogen peroxide (H_2_O_2_), primary antibody was added followed by incubation at 4 °C for 24 h. The sections that had been stored overnight were then washed and incubated with secondary antibody for 40 min at room temperature. At last, both DAB chromogen and hematoxylin were used to stain sections. The IHC results were evaluated independently by two pathologists. The protein expression score was defined as 0, 1, 2, or 3 according to the percentage of stained tumor cells (0–5%, 6–25%, 26–50%, and 50%–100%, respectively). The staining score was divided into low expression (0 and 1) or high expression (2 and 3).

### 2.12. Immunofluorescence

Cells were grown on glass slides, and after reaching 80% confluency, the medium was thrown away and cells were washed with PBS, and then fixed with 4% paraformaldehyde for 15 min. Subsequently, cells were blocked with goat serum for 30 min at 4 °C, followed by incubation with primary antibodies at 4 °C overnight. Cells were then incubated with fluorescence-conjugated secondary antibodies at room temperature for 1 h, followed by nuclear staining using DAPI. Images were acquired using a fluorescence microscope.

### 2.13. Xenograft Assay

Twenty male BALB/c nude mice aged 4 weeks old and weighing 18–20 g were purchased from Beijing HFK Bioscience Co. Ltd. (Beijing, China). All mice were placed under specific pathogen-free (SPF) conditions. All mice were divided into the pc-vector group and the pc-GATA5 group at random after 7 days of adaptation at the laboratory animal facility of Renmin Hospital of Wuhan University. For the tumor growth experiment, 1 × 10^6^ pc-vector or pc-GATA5 DU145 cells diluted in 200 μL of serum-free medium were injected into the forelimb axilla of the mice. The tumor formation status of the mice was measured and recorded every 5 days. After 35 days, the mice were sacrificed, and the tumors were dissected and photographed. The tumor volume (length × width^2^ × 0.5 mm^3^) was measured, and the tumor tissues were fixed in 4% paraformaldehyde for further analysis.

### 2.14. Statistical Analysis

SPSS Statistics 26 software was used for all statistical analyses. All data are expressed as the mean ± SD. Student’s *t* test was used for comparisons between different groups. The χ^2^ test was used to assess the correlation between the level of protein and clinical characteristics. Spearman’s correlation analysis was used to evaluate the relationship between target genes. *p* < 0.05 indicates statistical significance.

## 3. Results

### 3.1. GATA5 Is Downregulated in PCa

To clarify the function of GATA5 in PCa, we first analyzed its expression level in PCa samples and cell lines. The GEPIA database showed that the expression level of GATA5 was downregulated in PCa samples in contrast to normal samples (Figure 1A), and the TCGA database demonstrated that GATA5 expression in 499 PCa tissues was lower than normal samples (Figure 1B). Furthermore, analysis of 52 paired samples in TCGA also indicated that GATA5 was downregulated in PCa tissues in contrast with adjacent tumor tissues (Figure 1C). Receiver operating characteristic curve (ROC) analysis showed that GATA5 distinguished prostate cancer with an area under the curve (AUC) of 0.76, indicating that GATA5 may contribute to the diagnosis of PCa (Figure 1D). In addition, Western blotting (The whole western blot can be found in Appendix A) and qRT-PCR were used to assess GATA5 expression, which demonstrated that the GATA5 expression was lower in tumor samples than in tumor-adjacent samples (Figure 1E,F). In the PC3 and DU145 cell lines, the mRNA and protein of GATA5 were clearly lower than those in RWPE-1 cells (Figure 1G,H). We found that the methylation level of GATA5 promoter was higher in tumor tissues (Figure 1I), so the 5-Aza-CdR, a methyltransferase inhibitor, was treated in PCa cells to verify this hypothesis. As the results showed, the mRNA level of GATA5 was increased after being treated with 5-Aza-CdR. Overall, these data demonstrated that GATA5 is downregulated in both tumor tissues and cell lines.

### 3.2. Overexpression of GATA5 Inhibits PCa Cell Proliferation, Migration, and Invasion In Vitro

Based on the above analysis and experimental results, the effects of GATA5 on cell biological behavior were further explored in PC3 and DU145 cells. Western blotting analysis disclosed that the expression of GATA5 was higher in the pc-GATA5 group than the pc-vector group (Figure 2A). The CCK-8 assay demonstrated that the proliferative ability of cells with high GATA5 expression was significantly inhibited compared to pc-vector cells (Figure 2B). The colony formation and EdU assay results were consistent with the above results of the CCK-8 assay (Figure 2C–F). A wound-healing assay demonstrated that high expression of GATA5 attenuated migration (Figure 2G,H). In addition, a Transwell invasion assay was implemented and revealed that GATA5 decreased the number of cells invaded through the chamber (Figure 2I,J). Overall, these findings demonstrated that GATA5 overexpression inhibited the proliferation, migration, and invasion of PC3 and DU145 cells.

### 3.3. Effects of GATA5 on PCa Cell Cycle and Apoptosis

Flow cytometry was also utilized to explore the effect of GATA5 on the cell cycle and the cell apoptosis. In both PC3 and DU145 cell lines containing pc-GATA5, the percentage of cells in G2 phase clearly increased, while the proportion in G1 or S phase decreased (Figure 3A,B). Moreover, the proportion of apoptotic cells in the pc-GATA5 groups was higher compared with the pc-vector groups (Figure 3C,D). Western blotting analysis also showed that apoptosis-related proteins, such as Bax and Cleaved caspase-3, were upregulated in the pc-GATA5 group, whereas the level of Bcl-2 expression was significantly downregulated (Figure 3E). In summary, these results showed that GATA5 inhibits PCa progression by regulating the cell cycle and apoptosis.

### 3.4. GATA5 Mediates EMT via the FAK/PI3K/AKT Pathway

Enrichment of KEGG pathway analysis was used to explore the specific mechanism by which GATA5 affects prostate cancer progression through the cBioPortal database, and the results showed that GATA5 may regulate prostate cancer progression through the FAK pathway (Figure 4A). Western blotting was used to verify this hypothesis. In addition to p-FAK, the downstream pathways of FAK, such as p-PI3K and p-AKT, were significantly reduced (Figure 4B). Since phosphorylated FAK induces the expression of EMT markers, we investigated whether GATA5 affects the expression level of N-cadherin, vimentin, and E-cadherin. Western blotting analyses indicated that the EMT-associated genes, N-cadherin and vimentin, were downregulated in the pc-GATA5 group. In contrast, the level of E-cadherin was elevated. The Western blotting results were verified by immunofluorescence (Figure 4C,D). Overall, these data indicated that GATA5 inhibits EMT progression in PCa via the FAK/PI3K/AKT pathway.

### 3.5. GATA5 Suppresses PCa Cell Growth In Vivo

Next, to further study the function of GATA5 in the growth of prostate cancer cells, serum-free medium containing pc-vector or pc-GATA5 DU145 cells was injected into the forelimb axilla of the mice. As shown in Figure 5A,B, the tumor size and tumor growth rate in the pc-GATA5 group were significantly smaller and lower than those of the pc-vector group. In addition, IHC indicated that the expression of Ki67 protein was clearly downregulated in pc-GATA5 tumors compared to pc-vector tumors (Figure 5C,D). To further verify the above results, we found that the expression of Bax and Cleaved caspase-3 was higher in pc-GATA5 groups, while Bcl-2 had the opposite change (Figure 5E). Thus, these findings also suggested that GATA5 plays an inhibitory role in PCa progression in vivo.

### 3.6. GATA5 Inhibits PCa Cell Growth by Regulating PLAGL2 In Vitro

STRING analysis showed that GATA5 may interact with PLAGL2 to regulate PCa cell growth (Figure 6A). Western blotting analysis also disclosed that PLAGL2 was downregulated after transfection with pc-GATA5 in PCa cells (Figure 6B), and the immunoprecipitation results further confirmed that GATA5 could directly interact with PLAGL2 (Figure 6C). To further verify the role of PLAGL2 in PCa, the PLAGL2 knockdown cell line was constructed (Figure 6D). Cell proliferation experiments, including CCK-8, colony formation, and EdU assays, were then utilized to explore the mechanisms of GATA5 and PLAGL2. Similar to the role of GATA5, downregulation of PLAGL2 expression inhibited cell growth, and the strongest growth inhibitory effect resulted from both GATA5 overexpression and PLAGL2 knockdown (Figure 6E–I). Overall, these findings demonstrated that GATA5 inhibits prostate cancer progression by regulating PLAGL2.

### 3.7. GATA5 Participates in PCa Progression through Regulating PLAGL2 via the FAK/PI3K/AKT Pathway

Next, the wound-healing experiment and the Transwell invasion experiment were performed to verify whether GATA5 affects cell metastasis through PLAGL2. Overexpression of GATA5 and knockdown of PLAGL2 could reduce the migration ability of PCa cells, and the strongest migration inhibitory effect was disclosed in cells with both GATA5 overexpression and PLAGL2 knockdown (Figure 7A,B). The above results were consistent with the Transwell invasion assay (Figure 7C,D). In addition, transfection with pc-GATA5 and si-PLAGL2 reduced the levels of p-PI3K and p-AKT, accompanied by decreased p-FAK levels, and these levels were further reduced by co-transfection of pc-GATA5 and si-PLAGL2 (Figure 7E). In addition, the EMT was suppressed to a greater extent in the pc-GATA5 and si-PLAGL2 co-transfection group (Figure 7F). The expression of the apoptosis-related genes, Cleaved caspase-3 and Bax, was higher, while Bcl-2 expression was lower (Figure 7F). We wondered whether high expression of PLAGL2 could affect the biological function of PCa cells, and therefore, the PLAGL2 overexpression cell line was constructed (Figure 8A). We found that overexpression of PLAGL2 could alleviate the inhibitory effect of GATA5 on cell proliferation and metastasis (Figure 8B–J). In addition, PLAGL2 overexpression had the opposite function on the FAK/PI3K/AKT pathway compared with GATA5 (Figure 8K). Based on the above results, GATA5 may inhibit PCa progression through regulating PLAGL2 via the FAK/PI3K/AKT signaling pathway (Figure 9).

## 4. Discussion

It is estimated that PCa has the highest incidence of malignant tumors and is the second-ranked tumor-causing death of men in the USA [1]. Despite continuous advances in detection technology and treatment methods, the prognosis of PCa patients with metastasis is still poor [29]. Therefore, the search for new biomarkers of prostate cancer is especially urgent and vital. In our present study, we first indicated that GATA5 expression was lower in both PCa tissues and PCa cell lines compared to the adjacent tissues and RWPE-1 cells, respectively. High expression of GATA5 significantly retarded the progression of prostate cancer.

GATA5 belongs to the GATA gene family and is associated with a variety of diseases, including cancer. In hepatocellular carcinoma, it has been reported that GATA5 expression is lower due to aberrant hypermethylation, and the level of GATA5 mRNA is significantly associated with patients’ overall survival [19]. Research has also shown that GATA5 can suppress the tumor progression by downregulating the Wnt/β-catenin signaling pathway and inhibiting reprogramming genes [20]. In addition, GATA5 overexpression suppressed cholangiocarcinoma cells’ proliferation and metastasis through regulating the Wnt/β-catenin signaling pathway [30]. For colorectal and gastric cancer, GATA5 is silenced, and overexpression of GATA5 activates downstream antitumor target genes and can be used as a potentially valid noninvasive biomarker [15,16]. In addition, decreased GATA5 mRNA is closely correlated with recurrence-free survival and may serve as a biomarker for prognostic evaluation in kidney cancer [18]. A study has found that GATA5 may be associated with the risk of prostate cancer through genome-wide association studies [31]. Nevertheless, the role of GATA5 in PCa is still unclear. Our findings revealed that GATA5 was significantly under-expressed in prostate cancer by GEPIA and TCGA database analysis, which was consistent with our results in PCa samples and cells. Meanwhile, the mRNA level of GATA5 was upregulated after being treated with 5-Aza-CdR, a methyltransferase inhibitor, which indicated that GATA5 was hypermethylation in PCa cell lines. Moreover, we found that cell proliferation and metastatic ability were obviously inhibited in the pc-GATA5 group compared to the pc-vector group. Furthermore, GATA5 upregulation exacerbated cell apoptosis. Western blotting results demonstrated that the level of Cleaved caspase-3, an apoptosis-related protein, was upregulated by GATA5 overexpression, and the proportion of Bax/Bcl-2 was also elevated. Moreover, overexpression of GATA5 inhibited cell proliferation through inducing cell cycle arrest in G2 phase, and the proportion of cells in G1 or S phase also decreased.

PLAGL2, as an oncogene, contributes to the occurrence and progression of multiple tumors. In colorectal cancer, several studies have reported that PLAGL2 participates in colorectal cancer progression through activating the Wnt/β-catenin signaling pathway, which accelerates tumor growth and metastasis [32,33]. Furthermore, PLAGL2 promotes EMT via β-catenin-dependent regulation through ZEB1 [22]. For hepatocellular carcinoma, PLAGL2 weakens the effect of erlotinib, an anti-EGFR drug, through the EGFR-HIF-1/2α signaling pathway [34]. PLAGL2 has aberrant expression in bladder urothelial carcinoma, and is closely associated with clinical characteristics of the patients, such as tumor number, stage, and metastasis [35]. In a previous study, we had reported that PLAGL2 has high expression in both PCa tissues and cells, and the expression of PLAGL2 in metastatic PCa is higher than that in primary PCa. Additionally, PLAGL2 may serve as an independent factor for PCa prognosis and a potential biomarker for the diagnosis of PCa. However, the exact mechanism remained unclear. The analysis of STRING showed that GATA5 may interact with PLAGL2, and immunoprecipitation proved the above hypothesis. The present results showed that PLAGL2, which is negatively regulated by GATA5, promoted PCa cell proliferation and metastasis. Co-transfection of pc-GATA5 and si-PLAGL2 further reduced the OD value, number of colonies formed, migration capabilities, and invasion capabilities. Moreover, knockdown of PLAGL2 increased the number of apoptotic cells, and Western blotting analysis confirmed these results. Therefore, we wondered if there may be additional potential pathways regulated by GATA5, such as the Wnt/β-catenin signaling pathway. For further study, we constructed PLAGL2 high-expressing cell lines. As the results showed, overexpression of PLAGL2 could alleviate the inhibitory effect of GATA5 on cell proliferation and metastasis.

Enrichment of the KEGG pathway was used to explore the mechanisms by which GATA5 affected the progression of prostate cancer. The KEGG results demonstrated that GATA5 may be relevant to abnormal phosphorylation of FAK. PI3K and AKT are important downstream genes of FAK [36]. Western blotting showed that overexpression of GATA5 significantly inhibited the activity of FAK, and this effect was further strengthened after knocking down PLAGL2 at the same time. In addition, PLAGL2 overexpression had the opposite function on FAK compared with GATA5. PI3K/AKT, which are components of a vital signaling pathway in PCa, are downstream genes of FAK and are activated by FAK in the phosphorylated state. Therefore, we next evaluated the expression levels of PI3K/AKT in different groups of PCa cells, and these results were consistent with the above results.

EMT is the process by which tumor cells become more aggressive. Cells will discard some of the characteristics of normal cells but obtain mesenchymal features, resulting in more mobile and aggressive cells [37]. In the process of EMT, the expression level of E-cadherin, an epithelial marker, increases, but the levels of vimentin and N-cadherin are downregulated [38]. In PCa, there are many signaling pathways participating in EMT, and EMT plays an important role in tumor initiation, tumor progression, gain of stemness, tumor metastasis, and therapeutic resistance [39,40]. There have been many studies showing that the FAK signaling pathway promotes the EMT process of tumor cells [41,42]. We verified whether GATA5 affects EMT in PCa through the FAK signaling pathway. In the pc-GATA5 group, the results of Western blotting showed that the level of E-cadherin expression was increased when p-FAK expression decreased, while vimentin and N-cadherin were downregulated. Immunofluorescence was used to confirm the above results. At the same time, the expression levels of phosphorylated PI3K and AKT were consistent with those of p-FAK. In addition, PLAGL2 was knocked down in the pc-GATA5 group, which resulted in further increases or decreases in the above indicators. Next, we wondered if the results would be different if PLAGL2 was overexpressed, as the results showed that overexpression of PLAGL2 could attenuate the inhibitory effect of GATA5.

Finally, cells transfected with the pc-vector or pc-GATA5 were injected subcutaneously into nude mice to explore the effect of GATA5 in vivo. The result disclosed that GATA5 overexpression clearly promoted cell apoptosis and inhibited the growth rate of tumors, demonstrating that GATA5 inhibits tumorigenesis in vivo.

In conclusion, these findings demonstrated that GATA5 inhibits PCa progression in vitro and in vivo through negatively regulating PLAGL2 via the FAK/PI3K/AKT pathway. Our study disclosed for the first time that GATA5 plays a role in tumor-inhibiting effects on PCa. Overexpression of GATA5 decreases the proliferative activity of PCa cells via inducing cell apoptosis and cell cycle arrest, as well as by suppressing metastasis by inhibiting the EMT process. Thus, these findings suggest that GATA5 can serve as a new therapeutic target in the future.

## Figures and Tables

**Figure 1 cancers-14-02074-f001:**
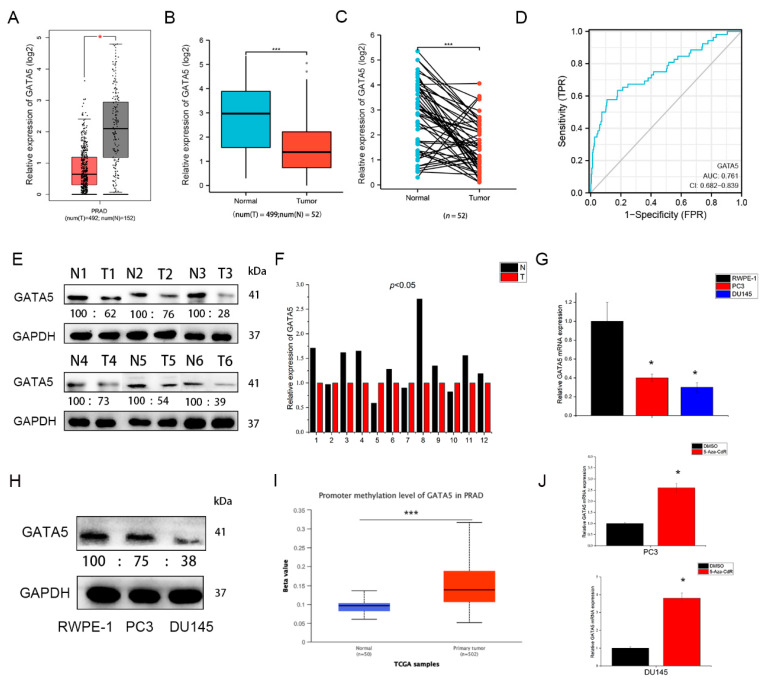
GATA5 is downregulated in both PCa samples and PCa cell lines. (**A**) GEPIA of GATA5 expression in 492 tumor and 152 normal tissues. (**B**) GATA5 expression was identified in 499 tumor and 52 normal samples from TCGA. (**C**) Expression in 52 pairs of tumorous and pericarcinomatous tissues from TCGA. (**D**) ROC analysis results of GATA5. (**E**) Western blotting showed the GATA5 expression in PCa and adjacent tumor tissues. (**F**) Relative mRNA expression of GATA5 in PCa and adjacent tumor tissues. (**G**,**H**) GATA5 was downregulated at both the mRNA and protein levels in PCa cell lines compared to RWPE-1 cells. (**I**) Promoter methylation level of GATA5 in PCa from TCGA. (**J**) The level of GATA5 mRNA in PCa cell lines after treatment with 5-Aza-CdR (5 μM) for 24 h. Data are presented as mean ± SD of at least three experiments. * *p* < 0.05 and *** *p* < 0.001.

**Figure 2 cancers-14-02074-f002:**
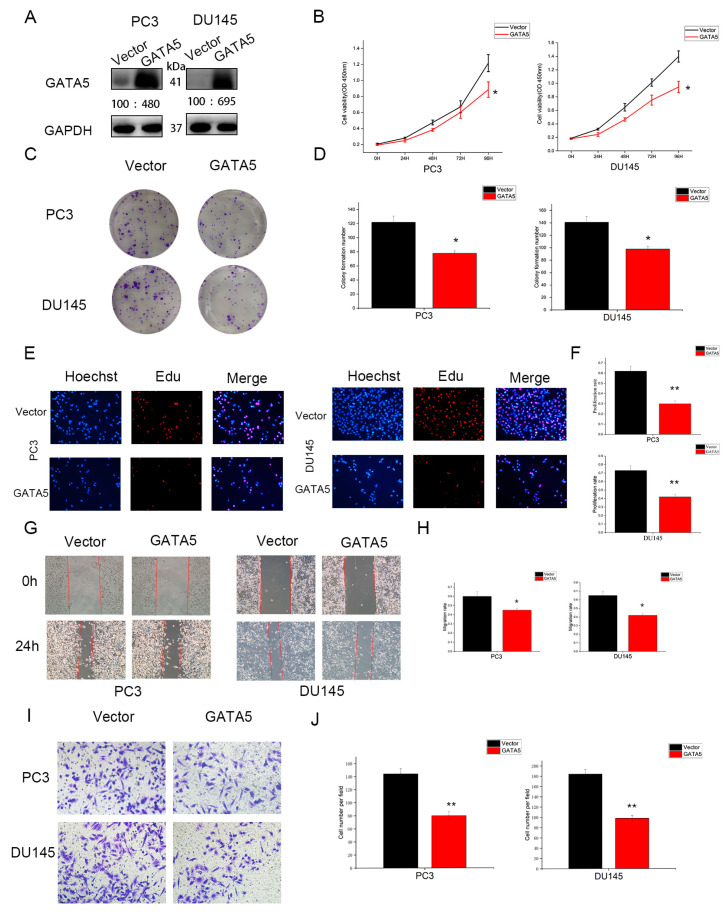
GATA5 overexpression inhibits the proliferation, migration, as well as invasion of PCa cells in vitro. (**A**) The effect of GATA5 overexpression was verified by Western blotting analysis. (**B**) Compared with the pc-vector group, the growth rate in the pc-GATA5 group was lower. (**C**,**D**) Overexpression of GATA5 suppressed colony formation. (**E**,**F**) The EdU assay showed that GATA5 overexpression decreased cell proliferation. (**G**,**H**) The migration capability was assessed through the wound-healing assay. (**I**,**J**) The invasive property was evaluated using a Transwell invasion assay. Data are presented as mean ± SD of at least three experiments. * *p* < 0.05 and ** *p* < 0.01.

**Figure 3 cancers-14-02074-f003:**
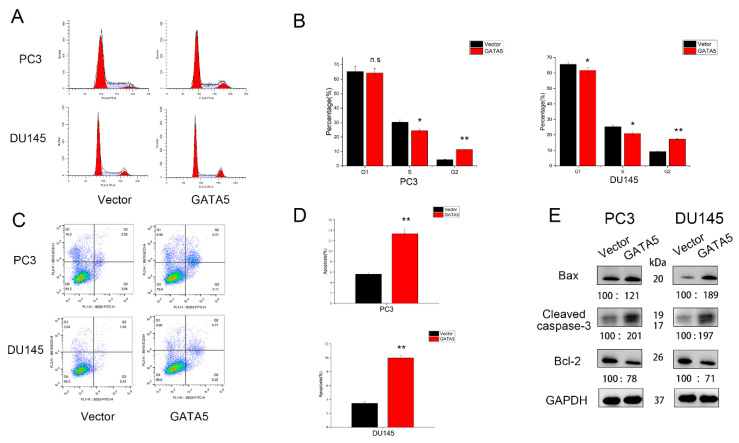
GATA5 overexpression promotes the cell cycle arrest and induces the apoptosis of PCa cells. (**A**,**B**) The results showed that GATA5 regulated the cell cycle. GATA5 overexpression elevated the G2 fraction of PCa cells. (**C**,**D**) The apoptotic rate in the pc-GATA5 groups was higher than in the pc-vector group. (**E**) The analysis of Western blotting clarified that GATA5 overexpression increased the apoptosis markers, Bax and Cleaved caspase-3, but decreased Bcl-2. Data are presented as mean ± SD of at least three experiments. * *p* < 0.05 and ** *p* < 0.01.

**Figure 4 cancers-14-02074-f004:**
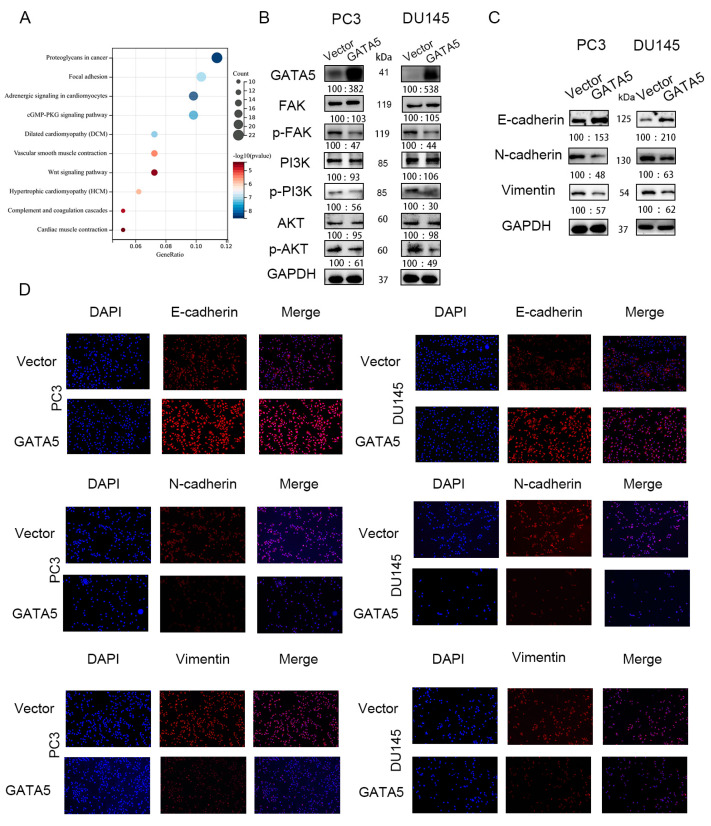
GATA5 mediates EMT via the FAK/PI3K/AKT pathway. (**A**) The result of enrichment of the KEGG pathway demonstrated that GATA5 was involved in PCa progression through the FAK/PI3K/AKT pathway. (**B**) The p-FAK expression and the downstream signaling pathway proteins, p-PI3K and p-AKT, were assessed by Western blotting analysis. (**C**) Overexpression of GATA5 inhibited the process of EMT, which was verified by the Western blotting. (**D**) Immunofluorescence verified the Western blotting results.

**Figure 5 cancers-14-02074-f005:**
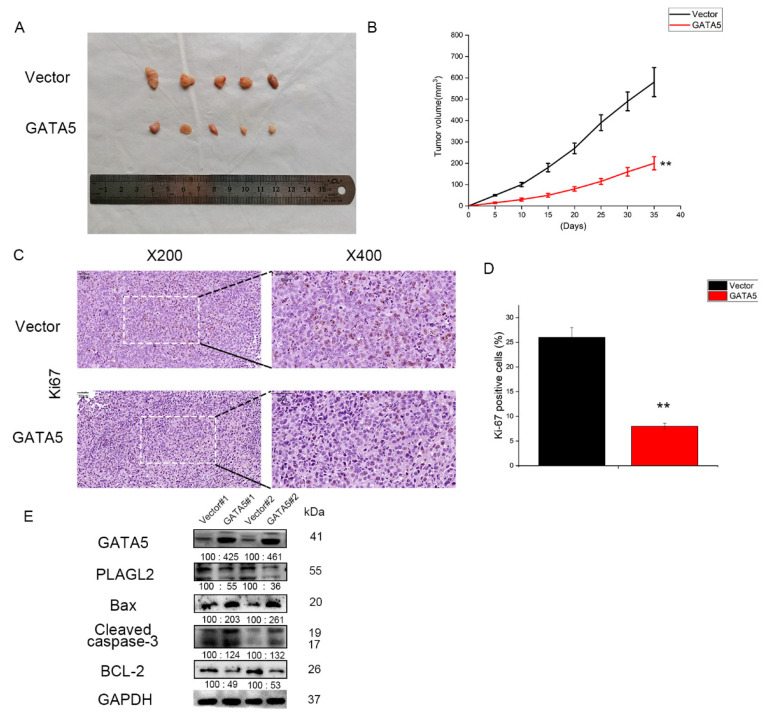
GATA5 overexpression suppresses PCa cell growth in vivo. (**A**) The tumor models in the pc-GATA5 group were much smaller when compared with the pc-vector group. (**B**) The tumors volume at each specific time point was calculated, and the results demonstrated that both the tumor growth rate and size in the pc-GATA5 group were smaller than those in the pc-vector group. (**C**,**D**) Immunohistochemistry was implemented to assess the expression of Ki67 in tumors. (**E**) Overexpression of GATA5 promoted prostate cancer cell apoptosis in vivo. Data are presented as mean ± SD of at least three experiments. ** *p* < 0.01.

**Figure 6 cancers-14-02074-f006:**
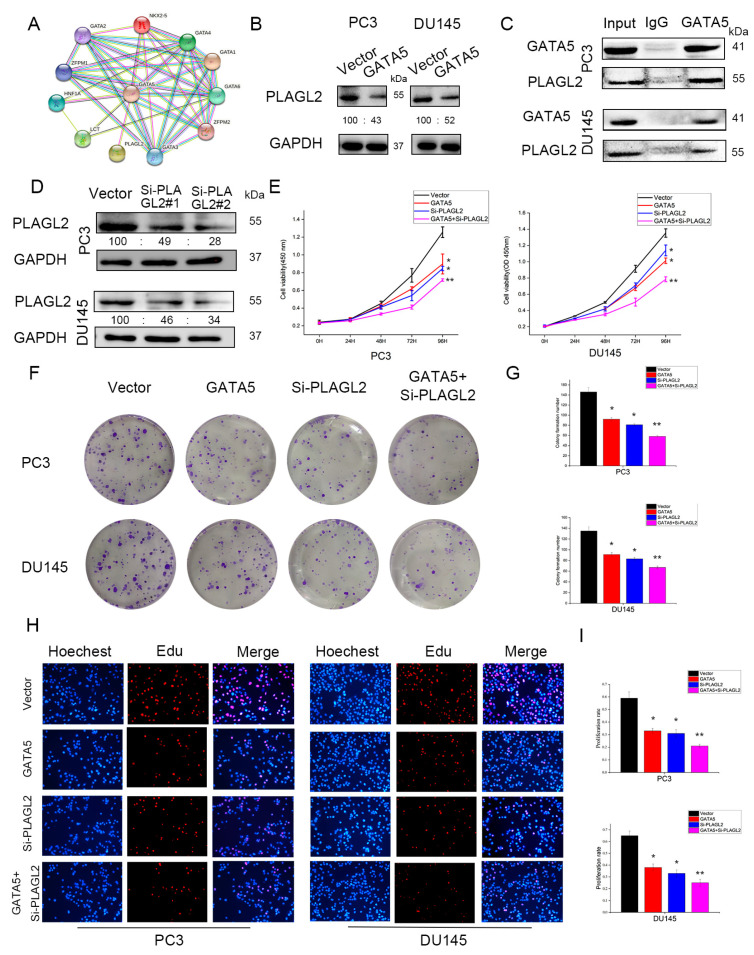
GATA5 inhibits PCa cell proliferation by regulating PLAGL2. (**A**) STRING analysis showed that GATA5 interacted with PLAGL2. (**B**) The expression of GATA5 was negatively correlated with PLAGL2 according to the Western blotting. (**C**) Co-immunoprecipitation indicated the interaction between GATA5 and PLAGL2. (**D**) The effect of PLAGL2 knockdown was verified by Western blotting. (**E**–**I**) The CCK-8 experiment, colony formation experiment, and EdU experiment were utilized to compare the viability in different groups of PC3 and DU145 cells. Data are presented as mean ± SD of at least three experiments. * *p* < 0.05 and ** *p* < 0.01.

**Figure 7 cancers-14-02074-f007:**
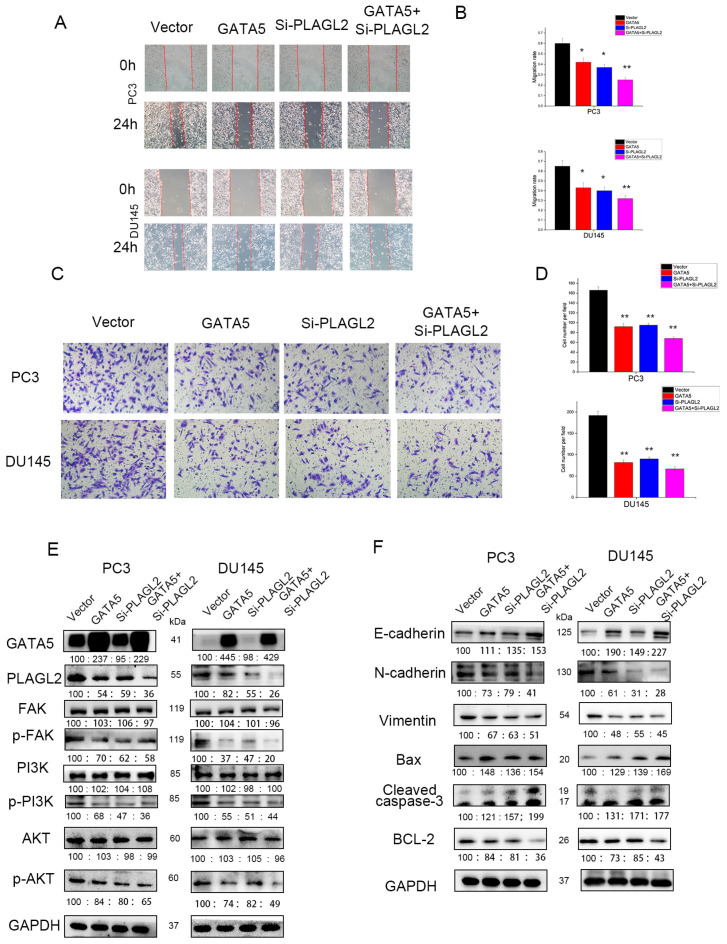
GATA5 participates in PCa progression through regulating PLAGL2 via the FAK/PI3K/AKT pathway. (**A**,**B**) A wound-healing experiment was utilized to explore the ability of migration in the four groups. (**C**,**D**) Transwell invasion assays demonstrated that knockdown of PLAGL2 in the pc-GATA5 group further reduced the number of cells that went through the chamber. (**E**) PLAGL2 knockdown enhanced the function of GATA5 overexpression in hindering the protein level of p-FAK, p-PI3K, as well as p-AKT. (**F**) GATA5 affected cell apoptosis and EMT by regulating PLAGL2. Data are presented as mean ± SD of at least three experiments. * *p* < 0.05 and ** *p* < 0.01.

**Figure 8 cancers-14-02074-f008:**
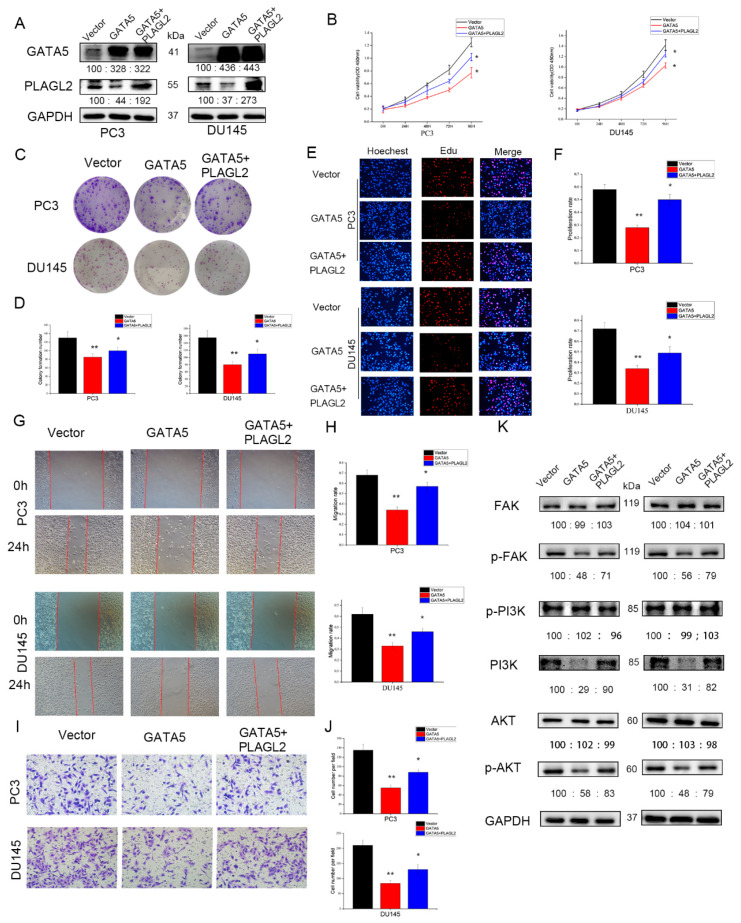
Overexpression of PLAGL2 reverses the effects of GATA5 in PCa cell lines. (**A**) The PLAGL2 overexpression cell line was constructed. (**B**–**F**) PLAGL2 overexpression could alleviate the inhibitory effect of GATA5 on cell proliferation. (**G**–**J**) The migration and invasion ability of cells were partially restored after overexpression of PLAGL2. (**K**) PLAGL2 overexpression had the opposite function on the FAK/PI3K/AKT pathway compared with GATA5. Data are presented as mean ± SD of at least three experiments. * *p* < 0.05 and ** *p* < 0.01.

**Figure 9 cancers-14-02074-f009:**
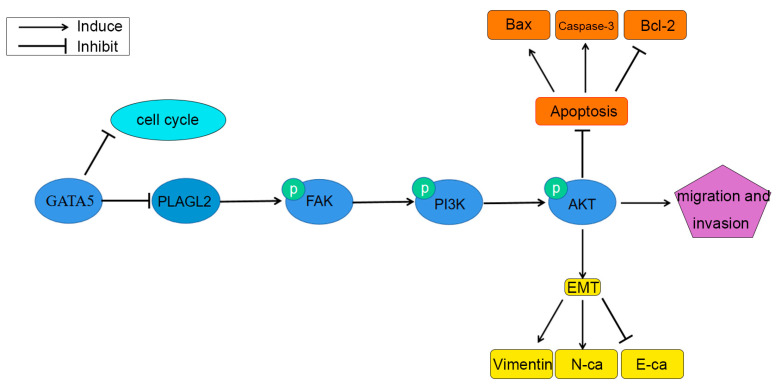
Overexpression of GATA5 inhibits prostate cancer progression by regulating PLAGL2 via the FAK/PI3K/AKT pathway.

**Table 1 cancers-14-02074-t001:** All the primary antibodies applied for the study.

		Dilution Rate				
Antibody	Specificity	WB	IF	IHC	IP	Company
GATA5	Rabbit	1:500	-	-	1:200	Proteintech
PLAGL2	Rabbit	1:500	-	-	1:100	Proteintech
E-cadherin	Rabbit	1:1000	1:100	-	-	Proteintech
N-cadherin	Rabbit	1:1000	1:100	-	-	Proteintech
Vimentin	Rabbit	1:1000	1:100	-	-	Proteintech
FAK	Rabbit	1:1000	-	-	-	ABclonal
p-FAK	Rabbit	1:1000	-	-	-	ABclonal
PI3K	Rabbit	1:1000	-	-	-	ABclonal
p-PI3K	Rabbit	1:1000	-	-	-	ABclonal
AKT	Mouse	1:1000	-	-	-	ABclonal
p-AKT	Mouse	1:1000	-	-	-	ABclonal
Bax	Rabbit	1:1000	-	-	-	CST
Bcl-2	Mouse	1:1000	-	-	-	CST
Cleaved caspase-3	Rabbit	1:1000	-	-	-	CST
Ki67	Rabbit	-	-	1:200	-	Abcam
GAPDH	Rabbit	1:1000	-	-	-	Servicebio

## Data Availability

The data presented in this study are availability in Appendix A here.

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
