# Peer review of "Overexpression of GATA5 Inhibits Prostate Cancer Progression by Regulating PLAGL2 via the FAK/PI3K/AKT Pathway"

_cancers, 2022, doi:10.3390/cancers14092074_

Round 1

Reviewer 1 Report

The authors have addressed my concerns adequately.
Please include the label for Fig 8K ( same as in Fig 8A).

This manuscript is a resubmission of an earlier submission. The following is a list of the peer review reports and author responses from that submission.

Round 1

Reviewer 1 Report

To authors

In this study, the authors showed that GATA5 downregulates PLAGL2 and inhibits PCa progression through suppression of FAK/PI3K/AKT pathway. Although the result of this study is interesting, there are some minor concerns.

  1. “GATA5 expression” label should be shown in Fig 1A-C.
  2. The number of samples is incorrect in Fig 1B (N=52, T=499).
  3. The label of the right graph should be shown in Fig 2B.
  4. The condition (control/GATA5) should be indicated in Fig 2C.
  5. The number of Edu positive cells should be counted and shown in Fig 2E and Fig 6G.
  6. IHC image is slightly dark in Fig4D.
  7. The number of Ki67 positive cells should be counted in Fig 5C.
  8. The authors should discuss more precisely how GATA5 regulates PLAGL2 and how PLAGL2 regulates FAK/PI3K/AKT pathway.

Reviewer 2 Report

In this manuscript, Wang et al., findings suggested that GATA5 downregulation is vital for the progression of prostate cancer. Using in-vitro and in-vivo studies, authors showed that overexpression of GATA5 decreases prostate cancer cell growth through apoptosis and decrease in EMT signaling. Additionally, authors showed that GATA5 regulates PLAGL2 via FAK/PI3K/AKT signaling to elicits its tumor suppressor role in prostate cancer.

The manuscript would benefit from the following corrections/clarifications.

  1. Authors mentioned - “Overexpression of GATA5 decreases the proliferative activity of PCa cells via inducing cell apoptosis and cycle arrest as well as by suppressing metastasis by inhibiting the EMT process.”

To prove that GATA5 is suppressing metastasis – Authors need to perform a proper in-vivo experiment and investigate lymph node metastasis/positive distant metastasis. They also need to investigate the regulation of various integrins by GATA5 for ex: ITGB1 and ITGA6 proteins contributes to prostate tumor metastasis to bone (PMID: 26239765). Unfortunately, the current in-vitro data on EMT and migration/invasion is not sufficient.

  1. As the authors pointed out in the manuscript, promoter hypermethylation may be one of the important mechanisms for the transcriptional inactivation of GATA5

Authors need to first establish this point, if GATA5 levels are decreased in prostate cancer cells due to the promoter hyper methylation or there is any unknown mechanism!

  1. There is an article published in 2015, PMID: 25658610 – suggests the connection of GATA5 deregulation in Prostate cancer – needs citation of this article in the current manuscript
  2. “Fig.5 - GATA5 overexpression suppresses PCa cell growth in vivo”

Authors should have investigated the expression of PLAGL2 and other apoptotic markers using western blot or Realtime-PCR in Vector tumors Vs GATA5 overexpressing tumors – to confirm their in-vitro findings

  1. Authors pointed out that – STRING analysis showed a possible interaction between GATA5 and PLAGL2 and inverse relation using StarBase dataset. First of all, the string analysis also showed several potential genes other than PLAGL2, moving forward- why authors picked PLAGL2 in the current study? Needs some solid experimental data. Secondly, Fig 6B data – the negative correlation doesn’t look very significant. Authors should have investigated PLAGL2 expression levels in the data sets from Fig 1, where they investigated GATA5 – to confirm the negative correlation.

Comments related to the existing Figures: First of all – Authors need to upload high quality figures – Very difficult to interpret the IHC, IF data using the current quality of figures

Figure 1.

  • Use full forms for GEPIA and TCGA and cite the reference for the three datasets used in the Figure 1. This data needs to be uploaded in the supplementary files as Excel sheet.
  • Y-axis label is missing for Figs 1A, 1B and 1C.
  • Fig 1B – Below x- axis, authors mentioned N=499 and T=52, whereas in the figure legend – they mentioned 499 tumors and 52 normal samples – which one is correct?
  • Fig 1E – Authors should have done a Real-time PCR on Normal Vs Tumor samples to check the mRNA expression of GATA5 – to confirm the results observed using public datasets, like they compared mRNA and Protein levels of GATA5 in Prostate cancer cell lines (RWPE1/PC3/DU-145). Please include quantification of western data – as its very difficult to appreciate the GATA5 protein difference in RWPE1 Vs PC3 cells.
  • No “n” number is provided in the figure legend for statistical purposes.

Figure 2.

  • Colony formation, 2C – Label the dishes like in Fig 6E
  • Please include treatment times for each experiment in the figure legend
  • In the methods section of Transwell Invasion Assay- “After the chambers were dried, the cells that migrated were counted under a microscope” – It should be cells that invaded! And how did authors count the cells? Used any software and how many fields were counted?
  • No “n” number is provided in the figure legend for statistical purposes.

Figure 3.

  • In Figure 3A – I can clearly see G2 increase in PC3 cells after GATA5 overexpression () but it’s very difficult to believe the G2 increase in DU145 cells.
  • In Figure 3B – Shows no change in G1 in fact authors mentioned n.s.(not significant). Whereas in the results, they said “ In both PC3 and DU145 cell lines containing pc-GATA5, the percentage of cells in G2 phase clearly increased, while the proportion in G1 or S phase all decreased (Fig. 3A-B)” and In the figure legend authors mentioned “GATA5 overexpression elevated the G2 fraction but decreased the G1 or S fraction ”, which is not visible from the figure! Please correct this!
  • Fig 3E – I believe authors meant cleaved caspase-3 – please change accordingly. Must include GATA5 blot.
  • No “n” number is provided in the figure legend for statistical purposes.

Figure 4.

Authors stated that - “Enrichment of KEEG pathway analysis was used to explore the specific mechanism by which GATA5 affects prostate cancer progression, and the results showed that GATA5 may regulate prostate cancer progression through the FAK pathway (Fig. 4A).”

  • What are your experimental groups for KEGG analysis– needs explanation. Is this analysis done using public datasets or just compared the PC3 or DU145 Ctr Vs GATA5 overexpression cells mRNA expression etc…and if datasets used for this analysis cite them properly
  • Fig 4B – Include GATA5 blot
  • Fig 4D – Immunofluorescence data – First of all, very poor-quality figure. It’s impossible to judge these figures using a physical print out- all figures look black. Used digital file to see these results- Number of cells (PC3 and DU145) in the E-cadherin/GATA5 image are quite high in comparison to Vimentin/GATA5 image – One would expect the equal cell number.

Figure 5.

Authors stated that “As shown in Fig. 5A-B, the tumor size and growth rate of the mice in the pc-GATA5 group were significantly greater than those of the pc-vector group”

  • As in Fig 5A-B - GATA5 group tumors size/volume is smaller as compared to pc-vector group – please correct!
  • It’s not the growth rate of the mice – it’s tumor growth
  • Methods section mentioned the use of 20 mice for this study – please explain how many mice used for each group and how many mice showed tumors.
  • No “n” number is provided in the figure legend for statistical purposes.

Figure 6.

  • Fig 6D/6E – Authors need to show PLAGL2 protein levels with/without Si-PLAGL2 transfection – to know the transfection efficiency of the siRNA. Must include two independent SiRNA’s targeting PLAGL2 for initial experiment. PLAGL2 inhibition has any effect on GATA5 expression levels?
  • Authors mentioned that – cell with GATA5 + SiPLAGL2 – showed strongest growth inhibition – Suggesting that GATA5 may suppress prostate cancer growth independent of PLAGL2. This needs to be discussed, like what are the additional potential pathways involved/regulated by GATA5.
  • Fig 6c – must include GATA5 blot
  • Authors should include a rescue experiment – Showing cells with increased PLAGL2 expression develops resistance to GATA5 overexpression

Figure 7.

  • Figure 7E and 7F – Include western blot quantification values – some of the changes are very low
  • Must include GATA5 blot – one needs to know if PLAGL2 has any effect on the GATA5 levels
  • Please include Total FAK and Total PI3K blots
  • Change Caspase 3 to cleaved caspase 3!
  • Western blot results in DU145 are more significant when compared to PC3 cells – even though the migration/invasion data looks pretty much similar- Any comments? Why this difference in the response?